# Effect of the Acidity Coefficient on the Properties of Molten Modified Blast Furnace Slag and Those of the Produced Slag Fibers

**DOI:** 10.3390/ma15093113

**Published:** 2022-04-25

**Authors:** Peipei Du, Yuzhu Zhang, Yue Long, Lei Xing

**Affiliations:** 1School of Metallurgy, Northeastern University, Shenyang 110819, China; 1710508@stu.neu.edu.cn; 2School of Metallurgy and Energy, North China University of Science and Technology, Tangshan 063009, China; zyz@ncst.edu.cn (Y.Z.); xlncst@126.com (L.X.)

**Keywords:** acidity coefficient, viscosity, crystallization, blast furnace slag, slag fibers

## Abstract

The online preparation of fibers using molten modified blast furnace slag can not only achieve the high-value-added utilization of the slag but can also make use of the sensible heat of the slag. In this paper, blast furnace slag was modified using iron tailings, and was then used to prepare slag fiber online; the effects of the acidity coefficient on the properties of the molten modified blast furnace slag and modified blast furnace slag fiber were investigated. With an increase in the acidity coefficient from 1.2 to 1.6, the temperature range of the slag melt, with viscosity in the 1–3 Pa·s range, increased from 101.2 °C to 119.9 °C. The melting temperature increased from 1326.2 °C to 1388.7 °C, and the suitable fiber-forming temperature range increased from 70.7 °C to 82.9 °C. With the increasing acidity coefficient, the crystallization temperature of the molten modified slag decreased markedly. When the acidity coefficient was greater than 1.4, the slag system was still in a disordered glassy phase at 1100 °C. The hardening speed gradually reduced with the increasing acidity coefficient when the modified slag was cooled at the critical cooling rate, resulting in a gradual increase in fiber formability. The fibers prepared from the modified slag at different acidity coefficients had smooth surfaces, and were arranged in a crossed manner at the macroscopic level. Their color was white, and small quantities of slag balls were doped inside the fibers. With an increase in the acidity coefficient from 1.2 to 1.6, the average fiber diameter increased from 4.2 μm to 8.2 μm, and their slag ball content increased from 0.73% to 4.49%. Overall, the acidity coefficient of modified blast furnace slag should be less than 1.5 in actual production.

## 1. Introduction

Slag fibers are filamentous inorganic fibers produced by centrifuging or injecting slag melt that is made primarily from industrial waste residues [1,2,3,4]. They have a specific chemical composition range: 36~42 wt.% SiO_2_, 28~47 wt.% CaO, 3~12 wt.% MgO, and 9~17 wt.% Al_2_O_3_ [5]. Slag fibers are lightweight and have low thermal conductivity, good chemical stability, and good sound absorption properties, and they are corrosion resistant. They can be used for thermal insulation, sound absorption and sound insulation material in buildings and thermal equipment, and they have high added value [6,7,8,9].

Blast furnace slag is solid waste formed during the process of ironmaking. The slag yield per ton of iron is 250~350 kg [10]. At present, blast furnace slag is mainly water-quenched slag. Water-quenched slag has good latent hydraulicity, and its utilization rate is 100% in the building materials industry [11]. However, when slag undergoes water quenching, it does not utilize the sensible heat of the blast furnace slag below 1500 °C, and its products have low added value [12,13]. Blast furnace slags have a chemical composition range of (36~42) wt.% SiO_2_, (38~49) wt.% CaO, (1~13) wt.% MgO and (6~17) wt.% Al_2_O_3_, and a small amount of sulfide [14,15]; in these premises, blast furnace slag melt should be a competitive candidate for the production of slag fiber. The online preparation of slag fibers from molten modified slag can not only realize a high added value of blast furnace slag but can also make use of the sensible heat from molten slag. The online preparation of slag fibers from molten modified blast furnace slag is a new process that can realize high added value and the large-scale utilization of blast furnace slag, thus providing a practical and effective approach for the utilization of blast furnace slag.

At present, many studies on the preparation of slag fibers from conditioned blast furnace slag have been conducted by domestic and foreign scholars. Li [16] studied the effects of MgO/Al_2_O_3_ on the viscosity–temperature characteristics and crystallization behaviors of conditioned blast furnace slag. The results showed that with an increase in the Mg/Al ratio, the viscosity of the conditioned blast furnace slag decreased, and its crystallization activation energy first increased and then decreased. Ren [17] used fly ash to modify blast furnace slag, and studied the performance changes during the modified process. The addition of fly ash could optimize the high-temperature viscosity and crystallization performance of blast furnace slag. Additionally, the crystallization temperature of the slag decreased with increasing fly ash addition, and the suitable fiber-forming temperature range first increased and then decreased. Li and Fan [18,19] studied the preparation of slag cotton by adding a silicon modifier to blast furnace slag, and found that the properties of the slag cotton met the requirements for the thermal insulation of external walls. All of these studies focused on the influence of individual factors on the performance of molten blast furnace slag or slag fibers. There are few reports that discuss the coupled effects of multiple factors on the properties of molten modified blast furnace slag and modified blast furnace fibers. The acidity factor is the mass ratio of acid oxides to alkaline oxides in melts, and it characterizes both slag modification and the effect of slag composition on fibers. Therefore, this paper thoroughly analyzes the effects of the acidity coefficient—*M_k_*, defined as the ratio of the wt.% SiO_2_ + wt.% Al_2_O_3_ to the wt.% CaO + wt.% MgO—on the viscosity and crystallization of molten modified blast furnace slag, as well as the diameter and the slag ball content of slag fibers, to provide a basis for the modification of blast furnace slag and the optimization of the fiber-forming process in the process for the preparation of slag fibers.

## 2. Materials and Methods

### 2.1. Experimental Raw Materials

In order to simulate the molten blast furnace slag environment, blast furnace slag was remelted in the experimental process. Dry blast furnace slag was the main experimental raw material, and iron tailings were the modified agent. In order to meet the requirement of the experimental equipment on the particle size of raw materials, after cooling naturally, the blast furnace slag was crushed to particles less than 3 mm in diameter. Iron tailings with an average particle size of 0.2 mm were directly taken from a tailings pit. The main chemical components of the experimental raw materials are shown in Table 1.

Table 1 shows that the SiO_2_ content of the blast furnace slag was small; this is consistent with the requirements of inorganic fibers in terms of raw material composition, as studied by Li [20]. The blast furnace slag was subjected to modification according to *M_k_*, the empirical assessment index of slag fiber composition stipulated by the GB/T11835-2016. For slag fibers, the *M_k_* should be less than 1.6, and the *M_k_* of blast furnace slag is 1.1, so the *M_k_* of modified blast furnace slag was taken as 1.2, 1.3, 1.4, 1.5 and 1.6. The chemical composition of modified blast furnace slag is shown in Table 2.

### 2.2. Viscosity Experiment

The temperature-dependent viscosity of the modified blast furnace slag at different *M_k_* values was measured using a melt physical property comprehensive tester (RTW-08, Northeastern University, China), and the viscosity constant was calibrated using castor oil, which is often used as the standard liquid for the calibration of blast furnace slag, at room temperature before the experiment [21]. A total of 140 g of the sample was weighed and placed in a dry graphite crucible. Then, the crucible was placed in the tester and heated in a furnace. The temperature of the furnace was maintained for 30 min after the temperature reached 1500 °C. Then, the viscosity as a function of the temperature was measured at a temperature variation rate of 3 °C/min. A schematic diagram of the experimental device is shown in Figure 1.

### 2.3. Crystallization Properties

The Equilb module and Phase Diagram module in FactSage thermodynamic software (FactSage 6.4, Montreal, QC, Canada) were used to simulate the crystallization behavior of the molten modified blast furnace slag during the experiment, and to obtain the theoretical crystallization temperature. The temperature range calculated in the simulation was from 1000 °C to 1500 °C. The crystallization behavior of the molten modified blast furnace slag melt was analyzed by a laboratory thermal simulation test. The blast furnace slag and iron tailings were manually proportioned and mixed according to the modified scheme, and a computer-controlled energy-saving high-temperature resistance furnace was used to remelt the modified blast furnace slag. The temperature was maintained for 30 min in order to ensure that the slag fully melted after the temperature reached 1500 °C, and then the slag was cooled to a specific temperature near the theoretical crystallization temperature within the furnace, and finally quenched. The microstructure and mineral phase composition of the water-quenched sample were analyzed using a scanning electron microscope (SEM, S-4800, Hitachi, Chiyoda-ku, Japan) and an X-ray diffraction (XRD, D/MAX2500PC, Science Co., Ltd., Tokyo, Japan) instrument, respectively. The particle sizes of the water-quenched slag samples in the XRD experiment were adjusted using a #200 mesh, the working current for the XRD was 40 mA, the working voltage was 45 kV, the scanning speed was 5°/min, and the scanning range, 2θ, was 10°–90°.

The crystallization behavior of molten modified blast furnace slags with different *M_k_* values during cooling at various rates was studied using a single hot thermocouple technique (SHTT, MTLQ-BQ-3, Chongqing University of science and technology, Chongqing, China). The cooling rates gradually increased from slow to fast, such that the critical cooling rate for the molten modified blast furnace slag for different *M_k_* values was deduced, and the continuous cooling transformation (CCT) curve of the conditioned blast furnace slag was constructed. The furnace temperature control system heated the furnace to 1500 °C at a rate of 5 °C/s, maintained the temperature for 300 s, and then cooled the furnace at a constant rate, as shown in Figure 2.

### 2.4. Centrifuging Fibrosis of Modified Blast Furnace Slag

Slag fibers were prepared by a centrifugation method on an experimental platform for the comprehensive utilization of metallurgical slag (independent research and development, North China University of Science and Technology, Tangshan, China). The experimental equipment included a 100 kg DC arc furnace, a four-roller centrifuge, and fiber collection equipment. The experimental equipment and a diagram of the fiber formation process are shown in Figure 3. During the process of fiber formation, 40 kg modified blast furnace slag was placed into the arc furnace, heated to 1500 °C, and kept at that temperature for 30 min in order to melt the slag uniformly. Then, the slag was slowly poured from the arc furnace into the four-roller centrifuge to form fibers. The speeds of the four rollers were 4060 r/min for roller No. 1, 4640 r/min for roller No. 2, 5220 r/min for roller No. 3, and 5800 r/min for roller No. 4. The inorganic fibers were collected in a fiber collection chamber.

### 2.5. Fiber Properties

An appropriate amount of the fibers was weighed and sorted into a small pile, and then bundles of equal length (1 cm) were cut from the middle of the pile and placed flat on a glass slide. Then, an appropriate amount of solution was added to the fibers, and the fibers were sorted evenly with a needle in order to ensure that the fibers were close to one another and arranged in a single layer. The width of 100 fibers was measured by optical microscopy (DM6000 M, Leica Co., Wetzlar, Germany), and the average diameter of the fibers was obtained by a simple calculation. The micromorphology of the fibers was observed by SEM. The fiber and slag ball were separated according to their buoyancy in water, and the slag ball content of the fiber was obtained by the corresponding calculation shown in Formula (1):(1)$$w=\frac{m}{m_{0}}$$
where *w* is the slag ball content (%), *m* is the mass of the slag ball with a particle diameter greater than 0.25 mm as fiber (g), and *m*_0_ is the mass of the fiber (g).

## 3. Results and Discussion

### 3.1. Viscosity of Molten Modified Blast Furnace Slag

Viscosity is an important performance factor for the use of molten modified blast furnace slag to prepare fibers. The viscosity range of the molten raw material used to prepare the slag fiber should generally be controlled within the range of 1–3 Pa·s [22]. Within the suitable fiber-forming viscosity range, a larger temperature range corresponding to the molten modified blast furnace slag melt means a wider fiber-forming temperature range and a more facile fiber-forming process. The viscosity–temperature curves of the molten modified blast furnace slag melt at different *M_k_* values are shown in Figure 4. The results in Figure 4 show that for all of the molten modified blast furnace slag with different *M_k_* values, the viscosity increases with a decreasing temperature. The viscosity–temperature curve for the molten modified blast furnace slag with an *M_k_* value of 1.2 shows an obvious inflection point; that is, the viscosity increases rapidly when the viscosity is greater than 2 Pa·s, showing short slag characteristics. The viscosity–temperature curves of the molten modified blast furnace slag corresponding to other *M_k_* values showed a gentler trend; that is, there were no obvious inflection points in the viscosity–temperature curves. With an increase in the *M_k_* value from 1.2 to 1.6, the temperature range of the molten modified blast furnace slag corresponding to the viscosity range of 1–3 Pa·s increased from 101.2 °C to 119.8 °C, as shown in Table 3. According to the basic structural theory of silicate melts [23], there are spaces between the larger [SiO_4_] groups of the molten modified blast furnace slag which can accommodate the interspersed movement of smaller SiO_4_ groups. When the temperature is high, there are more and larger spaces, which are conducive to the interspersed movement of small SiO_4_ groups, and the viscosity of the slag melt is low. When the temperature decreases, the spaces decrease, the movement of the SiO_4_ groups is blocked, the small SiO_4_ groups polymerize into large tetrahedral silica groups, and the viscosity of the slag increases. When the temperature remains unchanged, with an increase in the *M_k_* value, the SiO_2_ content in the modified blast furnace slag system increases, the mode of connection between SiO_4_ groups is simple, their arrangement is island-like, and they are closer together, resulting in an increase in the viscosity of the modified blast furnace slag. In addition, with an increase in the *M_k_* value, the ratio of Na_2_O to Al_2_O_3_ in the conditioned blast furnace slag system increases, and Al_3_^+^ captures more nonbridging oxygen atoms to form aluminum–oxygen tetrahedra, forming a unified network with the silicon–oxygen tetrahedra and resulting in the formation of complex aluminum-silicon-oxide anions [24,25]. This makes the slag structure more complex and inhibits the crystallization of the melt to a certain extent [26], resulting in an increase in the temperature range of the slag with viscosities in the between 1–3 Pa·s range.

For blast furnace slag, the fiber-forming process requires that the raw material melt has good fluidity; that is, it is in a free-flowing state. The minimum temperature at which slag can flow freely after melting is its melting temperature, T_m_. The melting temperature of molten modified blast furnace slag with different *M_k_* values can be obtained by making a 45° tangent line to its viscosity–temperature curve [27], as shown in Figure 5. The melting temperatures of the molten modified blast furnace slags increased with increasing *M_k_*, from 1326.2 °C when the *M_k_* value was 1.2 to 1388.7 °C when the *M_k_* value was 1.6, an increase of 62.5 °C. The melting temperatures of the molten modified blast furnace slags with different *M_k_* values were higher when the viscosity was 3 Pa·s and lower when the viscosity was 1 Pa·s. This is because, with the increase in *M_k_*, the content of the network-forming SiO_2_ oxides in the melt system increased, and they formed more complex anion groups that could activate and migrate to allow the slag to flow freely, thus requiring a higher temperature. During the fiber-forming process with modified blast furnace slag, the actual temperature of the slag must be higher than its melting temperature [28]. Therefore, the suitable fiber-forming temperature range for modified blast furnace slag with different *M_k_* values can be more accurately represented by the shaded area, T_s_, in Figure 5. The suitable fiber-forming temperature range of the modified blast furnace slag with different *M_k_* values is shown in Table 3. The table shows that with an increase in the *M_k_* value, the suitable fiber-forming temperature range gradually increases from 70.7 °C when *M_k_* is 1.2 to 82.9 °C when *M_k_* is 1.6, but the temperature ranges are all lower than the temperature ranges corresponding to the viscosity range of 1–3 Pa·s.

### 3.2. Crystallization Properties 

The process of fiber formation by blast furnace slag involves the rapid solidification of the slag from a melt with long-range disorder to a solid fiber state. This process must occur above the crystallization temperature of the slag melt. If crystallization occurs during the fiber formation process, then the fibers break, and the fiber performance deteriorates. The temperature at which crystallization begins for the molten modified blast furnace slag at different *M_k_* values during the cooling process was simulated using FactSage thermodynamic simulation software. When *M_k_* increased from 1.2 to 1.6, the initial crystallization temperatures of the molten modified blast furnace slag were 1398 °C, 1365 °C, 1328 °C, 1288 °C and 1257 °C. The molten modified blast furnace slags with different *M_k_* values were subjected to water quenching after furnace cooling to a specific temperature defined as the water-quenching temperature, and the phase composition of the water-quenched slag was analyzed using FactSage simulations. The water-quenching temperatures of the molten modified blast furnace slag at different *M_k_* values are shown in Table 4, and the phase analysis results are shown in Figure 6.

Figure 6 shows that when *M_k_* was 1.2 and the water-quenching temperatures were 1300 °C and 1400 °C, the XRD spectrum of the modified blast furnace slag was dispersed with a smooth bell-shaped peak without sharp diffraction peaks, indicating that no crystallization was occurring in the slag system. When *M_k_* was 1.2 and the water-quenching temperature was 1250 °C, some magnesium yellow feldspar and calcium aluminum yellow feldspar with lower peak strengths formed, as judged by the shape of the XRD spectrum; the results are consistent with the XRD results for molten slag reported by Ren [29]. When *M_k_* was 1.2 and the water-quenching temperature was 1200 °C, a large number of crystals began to appear in the slag system, as judged by the shape of the XRD spectrum. Slag fibers are an amorphous, inorganic, nonmetallic material. Therefore, the fiber-forming temperature should be above 1250 °C when the *M_k_* of modified blast furnace slag is 1.2. When *M_k_* was 1.3 and the water-quenching temperatures were 1350 °C and 1200 °C, the XRD patterns were dispersed, and exhibited smooth bell-shaped peaks without any signs of crystallization. When *M_k_* was 1.3 and the water-quenching temperature was 1150 °C, a large number of sharp diffraction peaks appeared in the XRD spectrum, and a large number of crystals—including magnesium yellow feldspar, calcium aluminum yellow feldspar, anorthite, wollastonite and pyroxene—precipitated from the slag system. When *M_k_* was 1.4, 1.5 and 1.6, and the water-quenching temperatures were 1100 °C, 1100 °C and 1050 °C, respectively, the XRD patterns were dispersed and had smooth bell-shaped peaks, indicating that no crystallization was occurring in the slag system. The viscosity of the slag melt increased with the decreasing temperature, the viscosity of this water-quenching temperature exceeded the maximum viscosity value in the appropriate fiber-forming viscosity range, and the slag melt did not easily form fibers. Therefore, when *M_k_* was greater than 1.3, the crystallization behavior of the slag melt was no longer the limiting factor determining slag fiber formation. Comparing the XRD spectra of the modified blast furnace slags with different *M_k_* values showed that the crystallization temperature of the slag system gradually decreased with an increasing *M_k_* value. When *M_k_* increased to a specific degree, it was difficult for crystals to precipitate in the molten modified blast furnace slag.

Generally, the composition of a slag system has a fundamental influence on its crystallization behavior. According to the theory of phase equilibria, the more complex the slag composition is, the lower the probability that each component collides and arranges into a certain ordered state from the disordered state during the slag cooling process to the liquidus temperature; that is, the lower the crystallization probability. Therefore, the XRD spectrum of a modified blast furnace slag system of a more complex composition and higher *M_k_* value shows crystallization at a lower temperature. In addition, the slag structure has an important influence on the crystallization behavior. With an increase in the *M_k_* value, the SiO_2_ content in the modified blast furnace slag system increases. The SiO_2_ in the slag melt is mostly present as special forms of silica tetrahedra or complex anions with large network connections, such as [SiO_4_]^4−^, [(Si_2_O_5_)^2−^]_x_ and [SiO_2_]_x_ [30], which makes it difficult for the melt to develop long-range ordered states during cooling, and means that the time required for crystallization increases. Therefore, with an increase in *M_k_*, the crystallization temperature for the molten modified blast furnace slag decreases.

Because the FactSage simulation was a thermodynamic equilibrium state simulation, the effects of kinetics, slag viscosity and other factors on slag crystallization were ignored. Therefore, the crystallization temperatures of the modified blast furnace slag obtained by the XRD analysis were lower than those of the FactSage simulation.

In order to analyze the crystallization behavior of the molten modified blast furnace slag with different *M_k_* values, the micromorphology of the water-quenched samples of the modified blast furnace slag with different *M_k_* values were evaluated by SEM, and the results are shown in Figure 7.

As shown in Figure 7, when *M_k_* was 1.2 and the water-quenching temperature was 1300 °C, the microstructure of the water-quenched sample was smooth without crystal precipitation. When the water-quenching temperature was reduced to 1250 °C, there was a small number of crystals of different shapes in the water-quenched sample. When the water-quenching temperature was reduced to 1200 °C, there were a large number of orderly, arranged crystals in the water-quenched sample, indicating that the modified slag had fully crystallized. When *M_k_* was 1.3 and the water-quenching temperature was 1150 °C, a large number of crystals precipitated from the water-quenched sample. When *M_k_* was increased to 1.4, 1.5 and 1.6, and the water-quenching temperatures were 1100 °C, 1050 °C and 1050 °C, there was no crystal precipitation in the water-quenched samples, and the micromorphology was a smooth glassy phase.

The melt cooling rate is the upper limit for the cooling rate that determines whether crystals precipitate during the slag cooling process. The upper limit of the cooling rate at which visible crystals cannot be precipitated in the melt is called the critical cooling rate. The CCT curves of modified blast furnace slag with different *M_k_* values are shown in Figure 8. When *M_k_* was 1.2, 1.3, 1.4 and 1.5, the critical cooling rates were 5 °C/s, 3 °C/s, 0.7 °C/s and 0.3 °C/s, respectively. When *M_k_* was 1.6, there was no crystal precipitation even when the cooling rate was less than 0.3 °C/s. The influence of the *M_k_* value of the modified blast furnace slag on its critical cooling rate is shown in Figure 9. Figure 9 shows that the critical cooling rate of the modified blast furnace slag decreased from 5 °C/s to 0.3 °C/s as *M_k_* increased from 1.2 to 1.5, and the reduction range gradually decreased. With an increase in *M_k_*, the SiO_2_ content in the modified blast furnace slag system increased, and the number of [SiO_4_]^4−^ ion clusters in the melt increased, which enhanced the interactions between the ion clusters, resulting in an increase in the viscosity of modified blast furnace slag, which increased the particle diffusion resistance, and the formation of short-range, ordered crystals required a longer crystal incubation time.

The process of the formation of slag fibers involves the transformation of the slag melt into products with a fixed geometry. During this process, the temperature of the slag melt must be within an appropriate temperature range above its crystallization temperature. At the same time, the influence of the medium around the slag on its hardening speed, Δη/Δt, should be considered. Figure 10 shows the corresponding hardening speed curves and viscosity–time curves for the modified blast furnace slag melt with different *M_k_* values cooled at its critical cooling rate. The Figure 10 shows that the steeper the slope of the tangent at a point on the curve is, the faster the hardening speed of the slag melt [31,32]. As shown in Figure 10, the hardening speed gradually decreased with increasing *M_k_* values when the modified blast furnace slag was cooled at its critical cooling rate, resulting in a gradual reduction in the thermal diffusivity and a gradual increase in fiber formability.

### 3.3. Performance Analysis of the Slag Fibers 

The appearances of the slag fibers prepared from the modified blast furnace slag at different *M_k_* values are shown in Figure 11. Figure 11 shows that the slag fibers had smooth surfaces and were arranged in a crossed manner at the macroscopic level; their color was white, and small quantities of slag balls were doped inside the fibers. The average diameters and slag ball contents of the slag fibers prepared from the modified slags with different *M_k_* values are shown in Figure 12. Figure 12 shows that with the increase in *M_k_* from 1.2 to 1.6, the average fiber diameter increased from 4.2 μm to 8.2 μm. When *M_k_* was less than 1.5, the average fiber diameter was less than 6 µm, which met the requirements of GB/t11835-2016. With an increase in *M_k_* from 1.2 to 1.6, the slag ball content increased from 0.73% to 4.49%; all of them were less than 7%, meeting the requirements of GB/t11835-2016. That is, the increase in the *M_k_* of the modified blast furnace slag simultaneously increased the average diameter and slag ball content of the slag fibers. 

## 4. Conclusions

(1) With an increase in the *M_k_* value of the modified blast furnace slag from 1.2 to 1.6, the viscosity of the slag melt increases continuously, the temperature range of the slag melt—with viscosity in the 1–3 Pa·s range—increases from 101.2 °C to 119.9 °C, the melting temperature increases from 1326.2 °C to 1388.7 °C, and the suitable fiber-forming temperature range increases from 70.7 °C to 82.9 °C.

(2) The crystallization temperature of the molten modified blast furnace slag gradually decreases with the increasing *M_k_* value. When *M_k_* is 1.2, there are crystals in the slag system at 1250 °C, and the crystals are mainly feldspar. When *M_k_* is greater than 1.4, the slag system has no crystals at 1100 °C.

(3) The critical cooling rate of the molten modified blast furnace slag decreases from 5 °C/s to 0.3 °C/s when *M_k_* increases from 1.2 to 1.5, and the reduction range gradually decreases. When *M_k_* is 1.6, there is no crystal precipitation even when the cooling rate is less than 0.3 °C/s. The hardening speed is gradually reduced with increasing *M_k_* values when the molten modified blast furnace slag is cooled at its critical cooling rate, resulting in a gradual amelioration in fiber improvement.

(4) The slag fibers prepared from modified blast furnace slags with different *M_k_* values have smooth surfaces, and are arranged in a crossed manner at the macroscopic level. Their color is white, and small quantities of slag balls are doped inside the fibers. With the increase in *M_k_* from 1.2 to 1.6, the average diameter of the slag fibers increases from 4.2 μm to 8.2 μm, and the slag ball content increases from 0.73% to 4.49%. With increasing *M_k_*_,_ both the average diameter and slag ball content of the slag fibers increase. 

(5) Based on the properties of quenched and tempered blast furnace slags with different *M_k_* values and the energy consumption of the process for the preparation of blast furnace slag fibers, the raw material system *M_k_* should be less than 1.5 in actual production.

## Figures and Tables

**Figure 1 materials-15-03113-f001:**
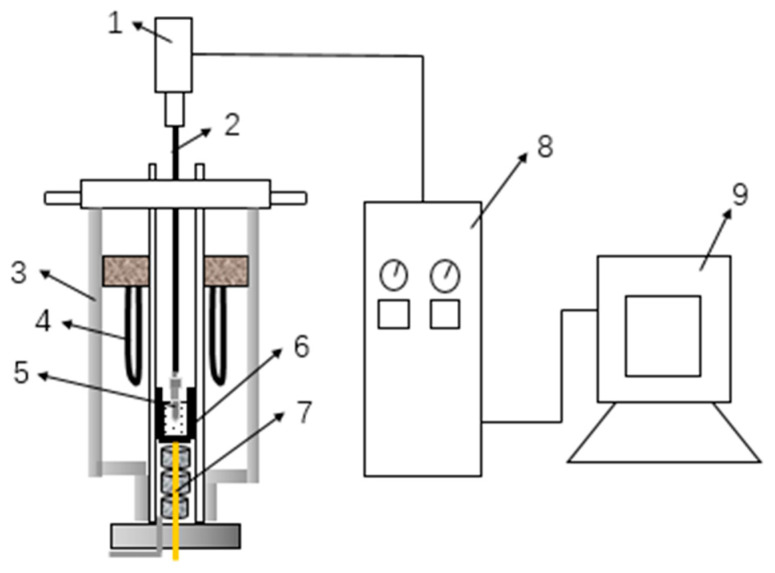
Schematic diagram of the viscosity measuring device. 1—Torque transducer; 2—corundum rod; 3—furnace body; 4—silicon-molybdenum heater; 5—molybdenum rotor; 6—graphite crucible; 7—thermocouple; 8—control cabinet and 9—computer.

**Figure 2 materials-15-03113-f002:**
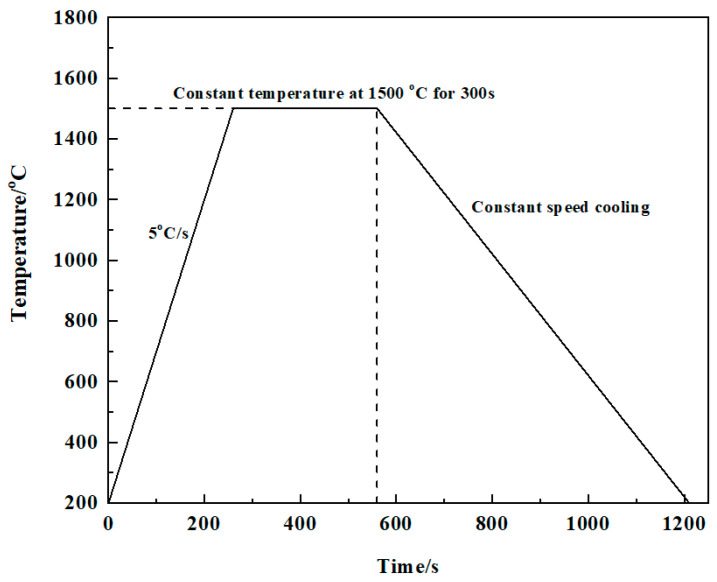
Schematic diagram of the temperature during the continuous cooling test.

**Figure 3 materials-15-03113-f003:**
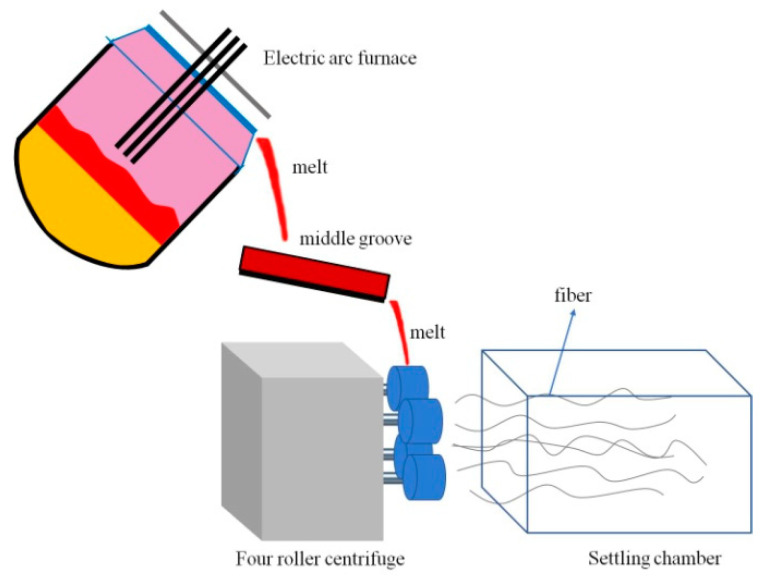
Schematic diagram of the experimental device and fiber forming process.

**Figure 4 materials-15-03113-f004:**
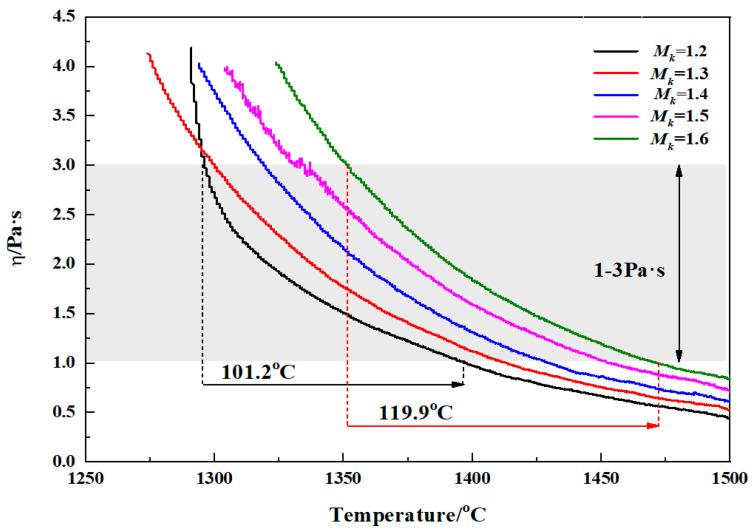
The viscosity–temperature curves of molten modified blast furnace slags with different *M_k_*.

**Figure 5 materials-15-03113-f005:**
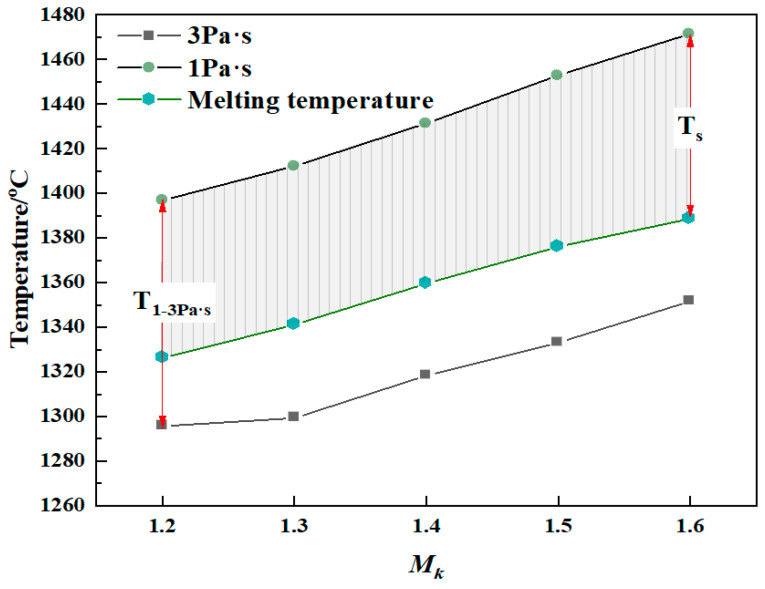
Effect of the *M_k_* value on the melting temperature of modified blast furnace slag.

**Figure 6 materials-15-03113-f006:**
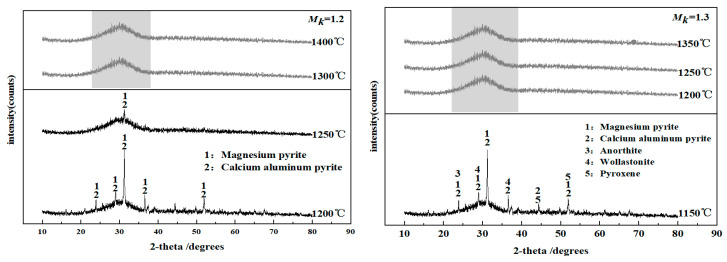
Effect of the final temperature during furnace cooling and the *M_k_* values on the phase composition of the modified blast furnace slag.

**Figure 7 materials-15-03113-f007:**
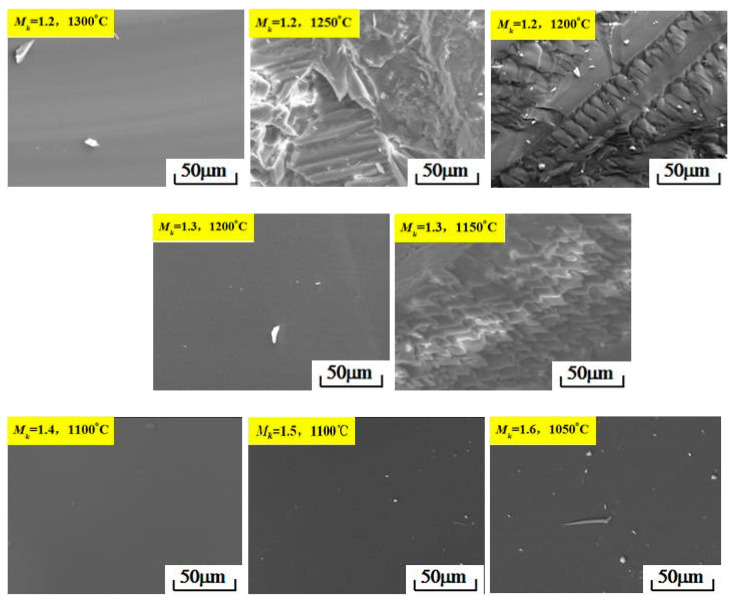
SEM images of the modified blast furnace slag with different *M_k_* values.

**Figure 8 materials-15-03113-f008:**
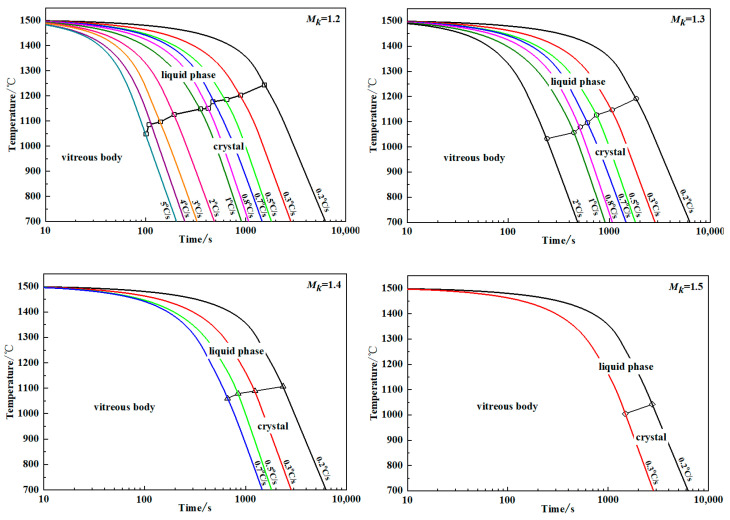
CCT curves of modified blast furnace slags with different *M_k_* values.

**Figure 9 materials-15-03113-f009:**
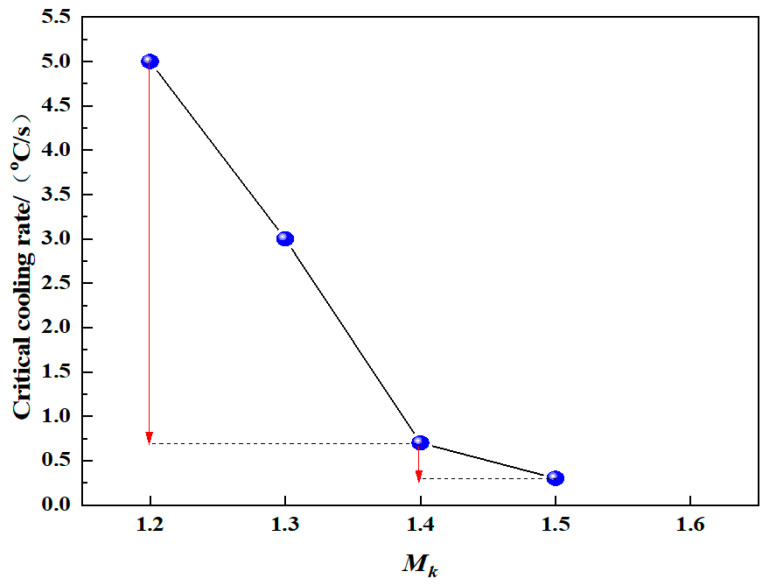
The influence of the *M_k_* value of the modified blast furnace slag on its critical cooling rate.

**Figure 10 materials-15-03113-f010:**
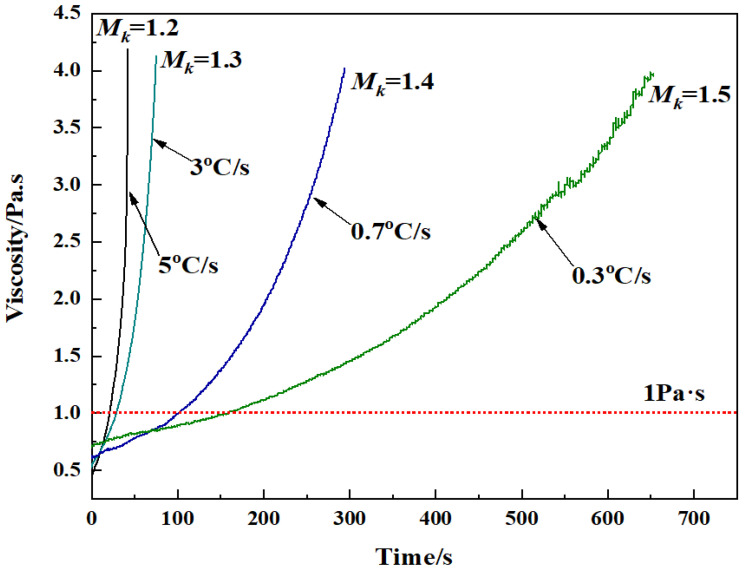
The corresponding hardening speed curves and viscosity–time curves for the modified blast furnace slag melt with different *M_k_* values cooled at its critical cooling rate.

**Figure 11 materials-15-03113-f011:**
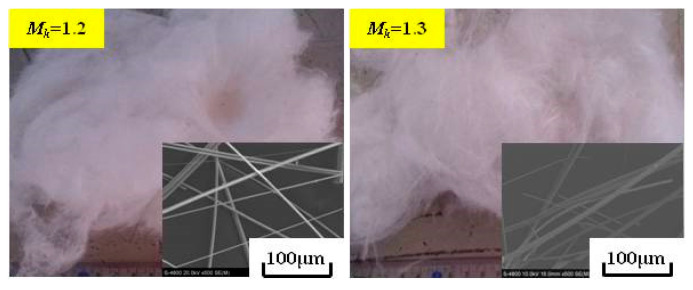
The appearance of the slag fibers prepared from the modified blast furnace slags with different *M_k_* values.

**Figure 12 materials-15-03113-f012:**
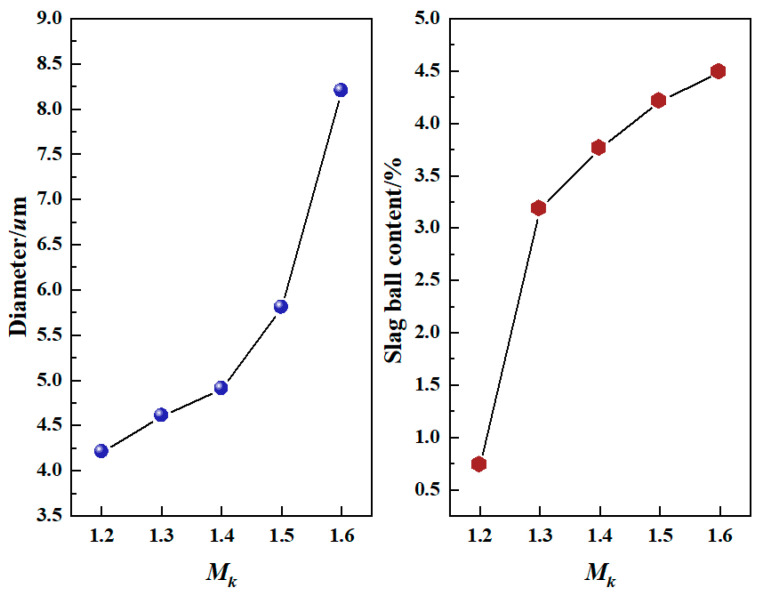
The average diameters and slag ball contents of the slag fibers prepared from modified blast furnace slags with different *M_k_* values.

**Table 1 materials-15-03113-t001:** Main chemical components of the raw materials/wt.%.

Raw Material	SiO_2_	CaO	MgO	Al_2_O_3_	TiO_2_	K_2_O	Na_2_O	Fe_2_O_3_	FeO
Blast furnace slag	32.60	36.43	8.72	15.44	1.55	0.71	0.55	0.07	0.46
Iron tailings	67.41	2.78	2.60	12.13	0.31	4.25	2.89	1.66	4.42

**Table 2 materials-15-03113-t002:** Chemical composition of modified blast furnace slag/wt.%.

Number	*M_k_*	Composition of Modified Slag								
		SiO_2_	CaO	MgO	Al_2_O_3_	TiO_2_	K_2_O	Na_2_O	Fe_2_O_3_	FeO
No.1	1.2	35.32	33.81	8.24	15.18	1.45	0.99	0.73	0.20	0.77
No.2	1.3	37.06	32.12	7.94	15.02	1.39	1.16	0.85	0.28	0.97
No.3	1.4	38.66	30.57	7.66	14.86	1.33	1.33	0.96	0.35	1.15
No.4	1.5	40.12	29.16	7.40	14.73	1.28	1.47	1.06	0.42	1.32
No.5	1.6	41.44	27.88	7.17	14.60	1.24	1.61	1.14	0.48	1.47

**Table 3 materials-15-03113-t003:** Suitable fiber-forming temperature range of modified blast furnace slag in the viscosity range of 1–3 Pa·s/°C.

*M_k_*	T_3Pa·S_	T_1 Pa·s_	T_1–3 Pa·s_	T_m_	T_s_
1.2	1295.7	1396.9	101.2	1326.2	70.7
1.3	1299.3	1412.2	112.9	1341.1	71.1
1.4	1318.3	1431.4	113.1	1359.6	71.8
1.5	1333.1	1452.9	119.8	1376.1	76.8
1.6	1351.7	1471.6	119.9	1388.7	82.9

**Table 4 materials-15-03113-t004:** The water-quenching temperature of molten modified blast furnace slag with different *M_k_*/°C.

*M_k_*	The Water Quenching Temperature of Molten Modified Blast Furnace Slag			
1.2	1400	1300	1250	1200
1.3	1350	1250	1200	1150
1.4	1300	1200	1150	1100
1.5	1250	1150	1100	1050
1.6	1250	1150	1100	1050

## Data Availability

Data sharing is not applicable.

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
