# Peer review of "Effect of the Acidity Coefficient on the Properties of Molten Modified Blast Furnace Slag and Those of the Produced Slag Fibers"

_materials, 2022, doi:10.3390/ma15093113_

Round 1
Reviewer 1 Report
The paper proposes an experimental study on the effects of acidity coefficient on the properties of the melt and slag fibers formed from blast furnace slag. Some contradictory statements are unclear, so the manuscript needs major revision and improvement.
- The title can be revised, and avoid abbreviations.
- Acidity coefficient (Mk) should be explained in the abstract before using an abbreviation.
- Abstract: the abstract needs to be summarised the main points and avoid unnecessary parts to better understanding and readability.
-Please underscore the scientific value added in the abstract. Add some of the most critical quantitative results to the Abstract.
- Keywords:
-Keywords must indicate the main materials, tests, and methodology used in the study. However, revising the keywords and writing based on the points mentioned above is required.
Revise as: acidity coefficient; viscosity; crystallization; blast furnace slag; slag fiber.
- The introduction section needs to improve.
It is essential to critically review the process of slag fibre production and the properties and potential application of these fibres and review how these fibres could use and enhance the performance of final products, and find out the gap in the study.
-Line 50: explain in more detail the specific chemical composition of slag fibers.
-Line 67: the various methods of preparation of slag fibers from conditioned blast furnace slag must be critically reviewed and find out the gap of study.
Lines 72 and 73: acidity coefficient (Mk) should be explained in more details and also the terms in the formula must be define and explain.
The study's objective is not clear and needs to be specified in the introduction section.
-All paragraphs should be developed with correct logic; the current version was seemed to be cobbled together.
- Materials and methods:
-The methodology must follow the standard specifications and cite them in the manuscript.
-The methodology must be written in more detail for better understanding.
-Before using any abbreviation, the full term must be explained in the text.
-Fig. 2: Check the unit in the y axis. Also, correct the temperature 1500 °C.
-Improve the Fig. 3 for better visibility.
- Results and discussion:
The results should be explained in more detail to better understand readers and compare the findings with the existing literature.
- Please check the legends of figures as some are not clear. Fig. 4 and 5, check the temperature unit.
- Table 3: write the unit.
- In Section 3.1: The relations between Mk, viscosity and temperature should be explained in more details and justify the reasons that increase in the Mk leads to rise in temperature and viscosity with a logic statements.
-Lines 185-187: “When the temperature remains unchanged, with an increase in the Mk value, the SiO2 content in the conditioned blast furnace slag system increases”. Justify the reason with logic statement.
-Fig. 6: revise the figure and clearly show the peaks.
-the XRD results must be explained in more details and justify the formation of each elements at peaks.
-Line 266: Explain in more details why with increase in Mk the crystallization reduced.
-In Section 3.3, I recommend adding more details on the physical properties of obtained slag fibres, such as strength and elongation.
- Conclusions
The conclusions need to be revised and improved. Please make sure the conclusion section underscores the scientific value added to the paper and the applicability of the findings/results. The conclusions are very general, with unnecessary statements.
Point 2: “When Mk is higher than 1.4, the slag system is still in a disordered glass phase at 1100 °C”; the statement is not clear, please revise and improve it.
General Comments:
- While, in science/engineering manuscripts, some minor mistakes in English can be understood before publication, this manuscript shows grammatical errors and typos. This can be seen as not also poor quality but also questionable intellectual merit of this paper. Thus, the readability of this manuscript is low, and the authors must have this paper proofread by a professional editor before the next submission.
- Please refer to more recent and relevant papers, as the references are not sufficient.
Reviewer 2 Report
Review of the article entitled “Effect of Mk on the properties of the melt and slag fibers formed from blast furnace slag” by Pei-pei Du, Yu-zhu Zhang, Yue Long, Lei Xing
The paper investigates the addition of iron tailings to the blast furnace slag for conditioning, and the effects of the Mk values of the conditioned blast furnace slag melt on its viscosity, crystallization behavior and conditioned blast furnace slag fiber properties. The paper shows a contribution from the prospect of using the blast furnace slag to prepare slag fibers. The following comments can be an improvement for the paper:-
- In the paper title I propose not to use the abbreviation Mk, please use the real wordings of the parameter (the acidity coefficient), and also try to show in the title that the iron tailings are used as a conditioning addition to adjust the chemical composition.
- In the introduction section, in addition to mentioning the 2021 production rates of slag in china, I propose to highlight the average tons of slag produced for each ton iron produced from the blast furnace to make the reader imagine the amount in worldwide scale.
- The introduction section also does not deeply cover the research related to the production of slag fiber from the blast furnace slag which is a main concern of the paper. The authors wrote “.. many studies on the preparation of slag fibers from conditioned blast furnace slag have been conducted by domestic and foreign scholars, but all of these studies have focused on….” But they did not mention any reference from these many. Therefore I recommend the last paragraph of the introduction section to be rewritten in a deeper way to elucidate what parameters have been studied, what were the outcomes, where is the gap?
- In the experimental section, the authors crushed the dry blast furnace slag to less than 3 mm, it will be more logical if they justified why to this size? Also it will be of value for the reader if they mentioned the exact open pit they get the iron tailing from. A justification also should be made why not using the hot slag directly?
- In the same section it is mentioned “Mk was taken as 1.2, 1.3, 1.4, 1.5 and 1.6” please justify why this range?
- Please correct the caption of figure 2 written as “Fig. 2 Schematic diagram of the viscosity measuring device”. Also the temperature unit is written as a question mark please correct.
- Figure 4, the formats of the temperature written numbers in the x-axis seem have some problems sometimes unnecessary spaces exist (please see 1300 and 1400). The units of the temperature range corresponding to a viscosity from 1-3 in the curves are incorrectly written.
- Figures 5 and 9, the units of the temperature at the y-axes should be corrected.
Reviewer 3 Report
The authors focus on the mixture of blast furnace slag and iron tailings, and the effect of acidity coefficient, Mk, on the properties of the slag melt and the slag fibers is discussed. The slag melt and slag fiber properties in an actual multi-component system are well summarized and it will contribute as a reference for slag utilization. I think this paper is worthy for publication after following minor points are properly fixed.
1. How the validity of viscosity-temperature curve is confirmed? This curve is base data to determine melting temperature and data to support the validity of the measurement is need.
2. In lines 281-284 (the fist paragraph in page 9), the noetwork structures of SiO_2 are discussed, but no supporting paper has been presented.
3. In line 338-341 (the first paragraph in page 11), it is written that “As shown in Fig. 10, the hardening speed gradually increased with increasing Mk values.” It seems that the slope of the tangent in viscosity-time curves becomes small with increasing Mk values. Please double check.
4. In figures 2, 6, 8, 9, and 10, some characters are garbled.
Round 2
Reviewer 1 Report
The authors have done the required corrections/improvements. However, check the formatting and structure of the paper before publication.
Author Response
Dear Reviewer and Editor,
Please see the attachment.
Thanks.

Reviewer 2 Report
Dear Authors
Thanks for the modifications and improvements you made in your manuscript.
The only couple of points I am still concerned about in your revised version are as follows:-
- The current revised title of the manuscript “Effect of Acidity Coefficient on properties of molten modifying blast furnace slag and modifying blast furnace slag fibers” is still poor and has repetition in wordings "modifying blast furnace slag". I propose to change it to be more informative one
A proposed title, and even you can improve it, can be: “Effect of Acidity Coefficient on the properties of modified molten blast furnace slag and those of the produced (formed, resulted) slag fibers”
- In many different places of the manuscript the word “modifying” is used to indicate that the blast furnace slag was modified, in my opinion this word should be replaced by the word “modified”
Apart from the above two mentioned points, I recommend accepting the paper for publication after minor changes.
Best wishes
Author Response

(The authors gave the same response as above.)
